# DriveDPO: Policy Learning via Safety DPO
# For End-to-End Autonomous Driving

**Shuyao Shang**[1,2] *     **Yuntao Chen**[3] *     **Yuqi Wang**[1,2]     **Yingyan Li**[1,2]     **Zhaoxiang Zhang**[1,2] †

[1]NLPR, Institute of Automation, Chinese Academy of Sciences,
[2]University of Chinese Academy of Sciences     [3]MiroMind

{shangshuyao2024, wangyuqi2020, liyingyan2021, zhaoxiang.zhang}@ia.ac.cn
chenyuntao08@gmail.com

## Abstract

End-to-end autonomous driving has substantially progressed by directly predicting future trajectories from raw perception inputs, which bypasses traditional modular pipelines. However, mainstream methods trained via imitation learning suffer from critical safety limitations, as they fail to distinguish between trajectories that appear human-like but are potentially unsafe. Some recent approaches attempt to address this by regressing multiple rule-driven scores but decoupling supervision from policy optimization, resulting in suboptimal performance. To tackle these challenges, we propose **DriveDPO**, a Safety Direct Preference Optimization Policy Learning framework. First, we distill a unified policy distribution from human imitation similarity and rule-based safety scores for direct policy optimization. Further, we introduce an iterative Direct Preference Optimization stage formulated as trajectory-level preference alignment. Extensive experiments on the NAVSIM benchmark demonstrate that DriveDPO achieves a new state-of-the-art PDMS of 90.0. Furthermore, qualitative results across diverse challenging scenarios highlight DriveDPO's ability to produce safer and more reliable driving behaviors.

## 1 Introduction

End-to-end autonomous driving has achieved remarkable progress in recent years. Unlike traditional modular pipelines, end-to-end methods directly predict future trajectories from raw sensor inputs, avoiding error accumulation across modules and simplifying the overall system design. Mainstream approaches [1–4] primarily rely on imitation learning, as shown in Fig. 1a. While effective in producing human-like behavior, imitation learning faces two critical safety issues. First, imitation learning minimizes the geometric distance between predicted and human trajectories. However, even slight deviations from human trajectories may lead to dangerous outcomes, as shown in Fig. 2a. Second, commonly used symmetric loss in imitation learning, such as mean squared error, penalizes deviations equally in both directions, while the safety impact can differ substantially across deviation directions, as shown in Fig. 2b. Consequently, policies trained solely through imitation often produce behaviors that appear reasonable but may be unsafe under actual driving conditions.

To address safety concerns, some recent methods [5–7] introduce rule-based teachers and adopt multi-target distillation to regress multiple rule-driven metrics as supervision signals, as shown in Fig. 2b. While such designs improve upon pure imitation learning regarding safety, they independently learn separate scoring functions for each anchor trajectory without directly optimizing the underlying policy distribution, ultimately leading to suboptimal driving performance.

---

*Co-first author.

†Correponding author.

39th Conference on Neural Information Processing Systems (NeurIPS 2025).

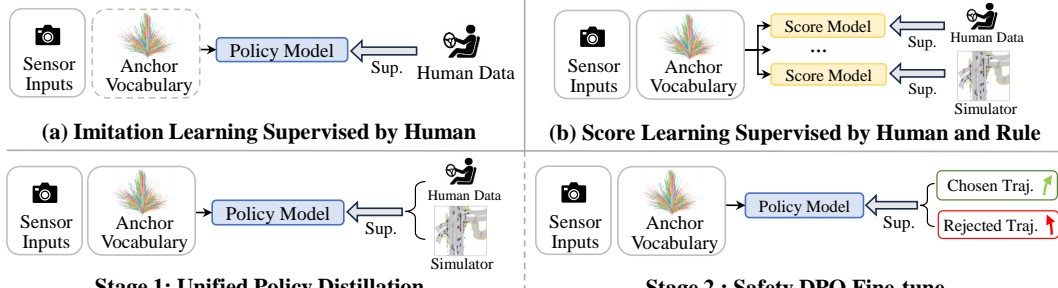

**(a) Imitation Learning Supervised by Human**

**(b) Score Learning Supervised by Human and Rule**

**Stage 1: Unified Policy Distillation**

**Stage 2 : Safety DPO Fine-tune**

**(c) Ous: Policy Learning via Safety DPO**

Figure 1: **Comparison of different training paradigms for end-to-end autonomous driving.** (a) The policy is trained by imitation learning. [3]. (b) The model trains multiple score heads using human and rule-based supervision signals [5, 6]. (c) Our method first pretrains the policy using a unified supervision signal that fuses human imitation similarity with rule-based safety scores, and then finetunes it via Safety DPO. Sup.: Supervision. Traj.: Trajectory.

Motivated by the safety issues of imitation learning and the indirect optimization limitation in score-based methods, we propose **DriveDPO**, a Safety Direct Preference Optimization Policy Learning framework. First, to address the challenge that score-based methods optimize per-anchor scores instead of the overall policy distribution, we propose a unified policy distillation approach that merges human imitation similarity and rule-based safety scores into a single supervisory signal for the policy model. Unlike score-based methods that construct separate score heads for each trajectory candidate, our method directly supervises the policy distribution over all anchors, enabling more coherent and end-to-end policy optimization. However, directly combining imitation and safety supervision into a single training objective formulates a multi-objective optimization problem [8–10]. To overcome this, we introduce the iterative Direct Preference Optimization framework [11] and propose Safety DPO, which reformulates the supervision as a trajectory-level preference alignment task. It enhances the policy's responsiveness to safety-oriented preferences through more stable and targeted optimization. Through preference learning on trajectory pairs, Safety DPO promotes human-like and safe trajectories over those that are human-like but unsafe, enabling more precise safety preference alignment in policy learning.

We evaluate the proposed framework on the NAVSIM benchmark [12] along with Bench2Drive benchmark [13]. Under a unified setting using a ResNet-34 perception backbone [14], our method achieves a new state-of-the-art PDMS of 90.0, surpassing the SOTA imitation-based method by 1.9 PDMS and the SOTA score-based method by 2.0 PDMS. Further qualitative results also show that our method substantially improves the safety of the learned policy in complex scenarios.

The contributions of this paper can be summarized as follows: **(1)** We identify fundamental challenges in existing imitation learning methods and score-based methods: Pure imitation learning fails to distinguish between trajectories that appear human-like but are potentially unsafe, while score-based methods decouple score prediction from direct policy optimization, resulting in suboptimal performance. **(2)** To overcome these challenges, we propose DriveDPO, a Safety DPO Policy Learning framework that first distills a unified policy distribution from human imitation and rule-based safety scores, followed by a DPO-based refinement stage for improved policy optimization. **(3)** We conduct comprehensive experiments on the NAVSIM benchmark and achieve a new state-of-the-art PDMS of 90.0, significantly advancing performance across multiple safety-critical metrics. By effectively suppressing unsafe behaviors, our method demonstrates great potential for safety-critical end-to-end autonomous driving applications.

## 2 Related Works

### 2.1 End-to-end Autonomous Driving

End-to-End Autonomous Driving [1–4, 15–23] typically maps raw sensor inputs to driving actions, either in the form of trajectories or low-level control commands, avoiding the cumulative errors and

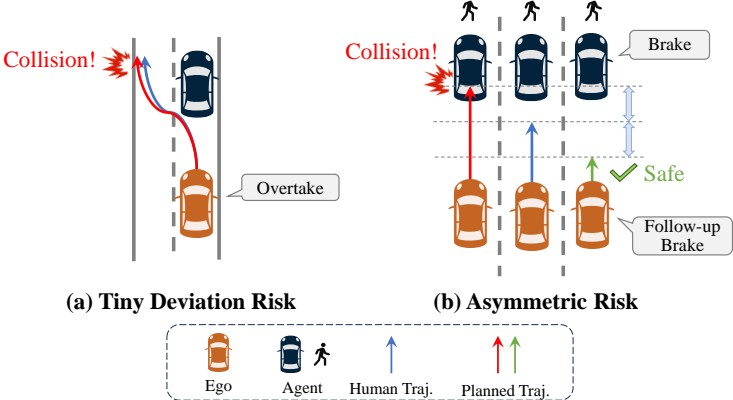

Figure 2: **Safety risks in imitation learning.** (a) In overtaking scenarios, even minor deviations from the human trajectory can lead to lane departure and collisions. However, imitation learning assigns low penalties to such trajectories due to their high similarity to human demonstrations, introducing serious safety risks. (b) In emergency braking scenarios, deviations in different directions have different safety implications: planned trajectories that overtake the human trajectory may cause rear-end collisions, while those that lag behind tend to be safe. However, imitation losses like mean squared error penalize both equally, failing to reflect the asymmetry of driving risks. Traj.: Trajectory.

interface bottlenecks inherent in traditional modular pipelines. UniAD [1] introduced a modular end-to-end architecture that enables a planning-oriented approach. VAD [2] further simplified UniAD's rasterized map representation into a vectorized format. VADv2 [3] introduced an anchor vocabulary, discretizing the continuous action space. SparseDrive [4] proposed an end-to-end method under the sparse paradigm. DiffusionDrive [24] incorporated a diffusion policy for trajectory prediction. These methods rely on imitation learning, which leads to critical safety issues. Recent works like Hydra-MDP [5] introduced a multi-teacher distillation framework that employs multiple score heads to regress imitation scores from human trajectories and rule-based metrics derived from simulator feedback. WOTE [7] employs a BEV world model to predict future BEV states of trajectories and scores them. However, these score-based methods lack direct policy distribution optimization, as they only optimize per-anchor scores independently. In contrast, we propose directly optimizing the policy distribution toward safety-aligned behavior.

## 2.2 Reinforcement Learning Fine-Tuning in Autonomous Driving

Reinforcement learning has gradually emerged as an important paradigm for autonomous driving research. [25] demonstrated an on-vehicle deep RL system for lane following using monocular input and distance-based reward. [26] introduced implicit affordances to enable model-free RL in urban settings with traffic light and obstacle handling. CIRL [27] combined goal-conditioned RL and human demonstration to improve success rates in CARLA. GRI [28] integrated expert data into off-policy RL for stable vision-based urban driving. Motivated by the success of reinforcement learning with human feedback (RLHF) [29–33] in large language models, an increasing number of studies have explored the paradigm of reinforcement learning finetuning in autonomous driving systems. DRIFT [34] proposed a Reward Finetuning strategy for unsupervised LiDAR object detection, where a heuristically designed reward function acts as a proxy for human feedback. BC-SAC [35] introduced using a pre-trained imitation policy as the initial policy and finetunes it in a simulated environment built from real driving data. Gen-Drive [36] proposed a behavior diffusion model that generates diverse candidate trajectories and is finetuned via reinforcement learning to favor higher-reward outputs. AlphaDrive [18] adopts a two-stage training strategy that combines supervised finetuning and reinforcement learning for planning-oriented reasoning. TrajHF [37], as a concurrent work, introduced a framework that finetunes a trajectory generator via reinforcement learning to produce trajectories more aligned with human driving style. However, TrajHF primarily focuses on driving style preference alignment without explicitly considering policy safety. In contrast, we introduce a safety RL finetuning into end-to-end autonomous driving, explicitly optimizing the policy to favor safer trajectories.

# 3 Preliminary

## 3.1 End to End Learning Task and Anchor Vocabulary

Given a raw sensor observation $O$, the model predicts trajectory points over the next $T$ time steps. Each trajectory point is represented as $(x_t, y_t, \theta_t)$, where $t = 1, \ldots, T$. Traditional methods typically perform continuous trajectory regression by predicting each point in continuous space. In contrast, we adopt the Anchor Vocabulary proposed in VADv2 [3], which transforms the action space from a continuous domain $\mathbb{R}^{T \times 3}$ to a predefined discrete set of anchors $\mathcal{V} = \{a^i \in \mathbb{R}^{T \times 3}\}_{i=1}^{N}$, where the size of the Anchor Vocabulary is $N$, $a^i$ denotes a trajectory consisting of $T$ consecutive points, each with position and heading $(x, y, \theta)$. Under this formulation, the policy model $\pi_\theta$ assigns a probability to each anchor in the vocabulary, forming a discrete action distribution:

$$\pi_\theta(a_i) = p_i, \quad i = 1, \ldots, N, \quad \sum_{i=1}^{N} p_i = 1. \tag{1}$$

The final predicted trajectory corresponds to the anchor with the highest probability:

$$a^* = \arg\max_{a_i \in \mathcal{V}} \pi_\theta(a_i) \tag{2}$$

## 3.2 Iterative DPO

Direct Preference Optimization (DPO) [38] is a preference optimization framework introduced for reinforcement learning fine-tuning of large language models [39–43]. The core idea of DPO is to optimize the policy directly based on preference pairs, encouraging it to favor preferred outputs over less preferred ones. To mitigate the out-of-distribution issues, iterative DPO [11] was proposed, which introduces an intermediate Reward Model to evaluate the quality of different outputs and has achieved notable success across various domains [44–48]. Formally, given a policy $\pi_\theta$ and a reward model $r_\phi$, for each input $x$ the policy generates a set of $N$ candidate trajectories $\{a_i\}_{i=1}^{N}$. The reward model assigns scalar scores $r_\phi(a_i)$ to each candidate. The sample with the highest score is selected as the chosen trajectory $a_w$, and the one with the lowest score is selected as the rejected trajectory $a_l$. The DPO loss then encourages the policy to prefer $a_w$ over $a_l$ relative to a fixed reference policy $\pi_{\text{Ref}}$:

$$\mathcal{L}_{\text{DPO}} = -\log \sigma \left( \beta \left( \log \frac{\pi_\theta(a_w)}{\pi_{\text{Ref}}(a_w)} - \log \frac{\pi_\theta(a_l)}{\pi_{\text{Ref}}(a_l)} \right) \right) \tag{3}$$

where $\sigma(\cdot)$ is the sigmoid function, and $\beta$ is a temperature parameter.

# 4 Method

In this section, we propose a Safety DPO Policy Learning framework. First, we introduce the overall policy model architecture (Sec. 4.1), which includes perception encoding, anchor tokenization, and cross-attention fusion for decision-making. To enable direct policy optimization that integrates human demonstrations and rule-based safety signals, we propose Unified Policy Distillation (Sec. 4.2), which integrates imitation similarity and rule-based score into a single supervision distribution. Finally, to alleviate the optimization challenges of multi-objective supervision and further improve policy safety, we present a Safety DPO method (Sec. 4.3), which fine-tunes the policy via iterative DPO.

## 4.1 Policy Model Architecture

The input to our model includes multi-view camera images, LiDAR point clouds, the current ego state, and a navigation command. The model outputs a probability distribution over a predefined set of $N$ discrete candidate trajectories, as shown in Fig. 3a. We begin by constructing a discrete set of anchor trajectories to define the output space of the policy network. Specifically, we apply $k$-means clustering on human driving trajectories from the NAVSIM Navtrain split [12] to obtain $N$ representative anchor trajectories, denoted as: $\mathcal{V} = \{a_i\}_{i=1}^{N} \in \mathbb{R}^{N \times T \times 3}$, $a_i \in \mathbb{R}^{T \times 3}$. Each anchor trajectory $a_i$ consists of $T$ future steps, with each step represented by $(x_t, y_t, \theta_t)$, $t = 1, \ldots T$. These anchors are passed through an Anchor Tokenizer, which begins by applying NeRF-style Fourier positional encoding [49] to capture spatial structure:

$$\Gamma = \gamma(\mathcal{V}) = \left( \sin(2^0 \pi \mathcal{V}), \cos(2^0 \pi \mathcal{V}), \ldots, \sin(2^{L-1} \pi \mathcal{V}), \cos(2^{L-1} \pi \mathcal{V}) \right) \in \mathbb{R}^{N \times T \times 6L} \tag{4}$$

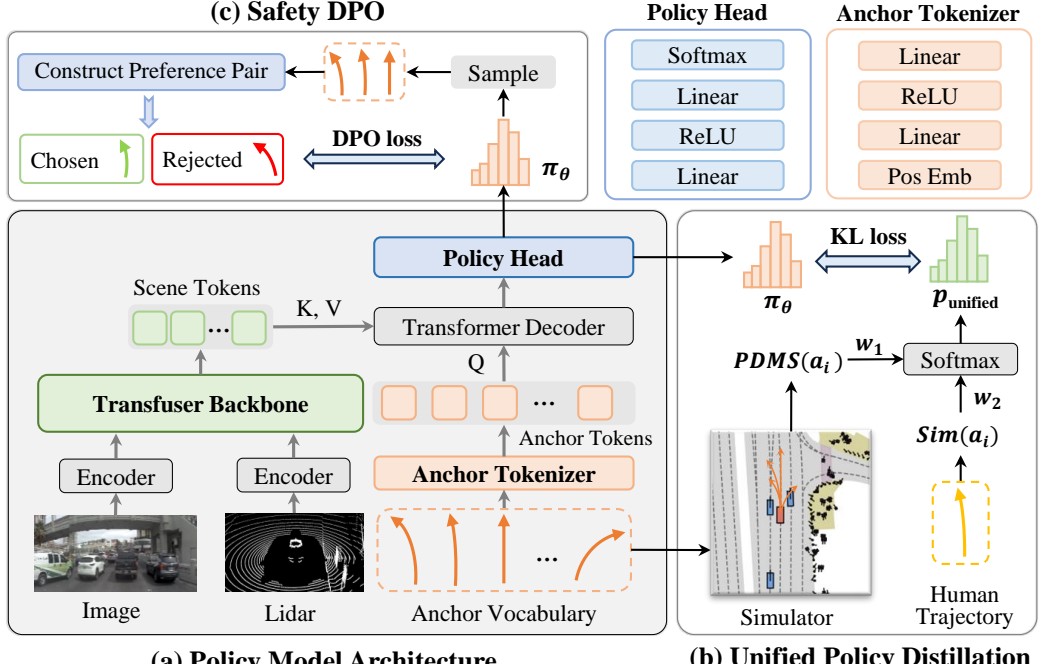

Figure 3: **Overall pipeline of the Safety DPO Policy Learning framework.** (a) Given multi-view camera images and LiDAR point clouds, a Transfuser backbone encodes the scene into a set of scene tokens. Meanwhile, a predefined Anchor Vocabulary is processed by an Anchor Tokenizer to produce anchor tokens. The anchor tokens attend to the scene tokens through a Transformer decoder to capture context-aware representations, which are then passed to the Policy Head to produce a probability distribution $\pi_\theta$. (b) To provide a unified supervisory signal, we compute a distribution $p_{\text{unified}}$ by combining human trajectory similarity and rule-based safety score. This distribution supervises the policy output via a KL divergence loss. (c) To further improve policy optimization, we introduce the iterative DPO framework. We sample $K$ candidate trajectories from the predicted policy distribution $\pi_\theta$, and construct a preference pair to fine-tune the policy via a DPO loss.

where $L$ is the positional encoding dimension. The encoded trajectories $\Gamma$ are then passed through a multi-layer perceptron (MLP) to produce the final anchor token representation:

$$E_{\text{anchor}} = \text{MLP}(\Gamma) \in \mathbb{R}^{N \times T \times d} \tag{5}$$

where $d$ is the embedding dimension. For perception, we adopt Transfuser [50] as the backbone to fuse camera images and LiDAR point clouds. The model first encodes the image and LiDAR inputs separately and then fuses them through multiple Transformer [51] modules to obtain a unified scene representation. This results in a set of Scene Tokens: $E_{\text{scene}} \in \mathbb{R}^{M \times d}$, where $M$ is the number of scene tokens. In the final decision-making stage, inspired by [2, 5, 6], we employ a Cross-Attention-based Transformer Decoder [51] to align anchor tokens with the scene context. Each anchor token acts as a query and attends over the scene tokens to obtain a context-enhanced representation:

$$\tilde{E}_{\text{anchor}} = \text{TransformerDecoder}(Q = E_{\text{anchor}}, K = E_{\text{scene}}, V = E_{\text{scene}}) \in \mathbb{R}^{N \times d} \tag{6}$$

where $\tilde{E}_{\text{anchor}}$ denotes the fused anchor representation enriched with scene semantics. Finally, the policy network applies an MLP to each context-enhanced anchor token $\tilde{E}_{\text{anchor}}$, followed by a softmax function to obtain the final policy distribution over anchor trajectories:

$$\pi_\theta(a_i) = \text{Softmax}(\text{MLP}(\tilde{E}_{\text{anchor}}))_i, \quad i = 1, \ldots, N \tag{7}$$

## 4.2 Unified Policy Distillation

Score learning methods [5–7] typically construct independent score heads for each trajectory anchor, predicting per-anchor scores rather than directly optimizing the policy distribution. To address this

issue, we propose a unified policy distillation approach that compresses all supervision signals into a single policy distribution target, enabling end-to-end consistency optimization of the policy output. Specifically, we combine human imitation similarity and a rule-based safety score to serve as a supervisory signal over the output probability distribution. For each candidate trajectory $a_i \in \mathbb{R}^{T \times 3}$ in the anchor vocabulary $\mathcal{V}$, we first define the imitation similarity between each candidate trajectory $a_i$ and the human reference trajectory $\hat{a} \in \mathbb{R}^{T \times 3}$ as the negative Euclidean distance. To ensure comparability across different scenes, we apply a softmax function to obtain the normalized relative similarity $\text{Sim}_i$ across all anchors:

$$\text{Sim}(a_i) = \text{Softmax}\left(-\|a_i - \hat{a}\|_2\right), \quad i = 1, \ldots, N \tag{8}$$

To obtain a rule-based safety score for each anchor trajectory, we use the high-fidelity NAVSIM simulator [12] which can perform forward simulation for each trajectory and returns multiple rule-based indicators, including No At-Fault Collision (NC), Drivable Area Compliance (DAC), Ego Progress (EP), Time-to-Collision (TTC), and Comfort (C). These metrics are then aggregated into a single scalar score known as the PDM Score (PDMS), computed as: $\text{PDMS} = \text{NC} \times \text{DAC} \times (5 \times \text{EP} + 5 \times \text{TTC} + 2 \times \text{C})/12$. We then can evaluate each anchor trajectory using the NAVSIM simulator to compute its corresponding PDMS through forward simulation:

$$\text{PDMS}(a_i) = \text{ForwardSimulation}(a_i), \quad i = 1, \ldots, N \tag{9}$$

In constructing the unified supervision distribution for policy learning, we introduce the log transformation to map the imitation similarity and the safety score from the range of $[0, 1]$ to $(-\infty, 0]$, which amplifies the differences when the values are small. As a result, if an anchor has a low safety score, its corresponding value after the transformation will be significantly lower than that of a safe anchor, effectively distinguishing safe trajectories from unsafe ones. In contrast, score-based methods regress scores for each anchor independently and cannot capture the sharp disparity between unsafe and safe trajectories. Moreover, the log transformation preserves the relative differences in imitation similarity, guiding the policy to favor those that are safe and more aligned with human behavior. Finally, we apply a softmax function to construct a soft-target distribution to introduce a competition mechanism among candidate anchors. The final unified supervision distribution $p_{\text{unified}}$ is:

$$p_{\text{unified}}(a_i) = \text{Softmax}\left(w_1 \cdot \log(\text{Sim}(a_i)) + w_2 \cdot \log(\text{PDMS}(a_i))\right), \quad i = 1, \ldots, N \tag{10}$$

where $w_1$ and $w_2$ are weighting coefficients. We train the policy model by minimizing the KL divergence between the predicted distribution $\pi_\theta$ and the unified supervision distribution $p_{\text{unified}}$:

$$\mathcal{L}_{\text{unified}} = \text{KL}\left(p_{\text{unified}} \,\|\, \pi_\theta\right) \tag{11}$$

### 4.3 Safety DPO

With the pre-trained policy obtained via unified policy distillation, a multi-objective optimization problem [8–10] arises due to the joint supervision from both imitation and safety signals. To mitigate this challenge, we adopt the iterative DPO framework [11] and propose Safety Direct Preference Optimization (Safety DPO), which further refines the policy by explicitly favoring trajectories that are both human-like and safe while suppressing those that appear human-like but risky. We begin by sampling $K$ candidate trajectories from the current policy distribution $\pi_\theta \in \mathbb{R}^N$. The naive approach selects the trajectory with the highest score in $p_{\text{unified}}$ as the chosen sample and the one with the lowest score as the rejected sample. However, this approach results in overly simplistic preference pairs, limiting the effectiveness of preference-based optimization. To address this, we continue to use the highest-scoring trajectory as the chosen sample $a_w$ but design two strategies for selecting the rejected sample $a_l$. The first selection method is Imitation-Based Rejected Trajectory Selection, which identifies trajectories that are spatially close to the human reference but exhibit poor safety performance. Specifically, it selects the rejected trajectory that is closest to the human trajectory while having a low PDMS:

$$a_l = \arg\min_{a_i} \|a_i - \hat{a}\|_2 \quad \text{s.t. PDMS}(a_i) < \tau, \quad i = 1, \ldots, K \tag{12}$$

where $\tau$ is a predefined safety threshold and $\hat{a}$ denotes the human trajectory. The second selection method is Distance-Based Rejected Trajectory Selection, which identifies unsafe candidates spatially close to the chosen trajectory. Specifically, it selects the rejected trajectory that has a low PDMS but is closest to the chosen trajectory $a_w$:

$$a_l = \arg\min_{a_i} \|a_i - a_w\|_2 \quad \text{s.t. PDMS}(a_i) < \tau, \quad i = 1, \ldots, K \tag{13}$$

Table 1: **Comparison of end-to-end driving methods on the NAVSIM.** The best results are denoted by **bold** and the second best results are denoted by underline. C: Camera. L: LiDAR.

| Method | Supervision | Input | NC↑ | DAC↑ | EP↑ | TTC↑ | C↑ | PDMS↑ |
|---|---|---|---|---|---|---|---|---|
| PDM-closed [52] | – | GT Perception | 94.6 | 99.8 | 89.9 | 86.9 | 99.9 | 89.1 |
| Human | – | – | 100.0 | 100.0 | 87.5 | 100.0 | 99.9 | 94.8 |
| Ego Status MLP | Human | Ego State | 93.0 | 77.3 | 62.8 | 83.6 | 100.0 | 65.6 |
| VADv2 [3] | Human | C | 97.9 | 91.7 | 77.6 | 92.9 | 100.0 | 83.0 |
| UniAD [1] | Human | C | 97.8 | 91.9 | 78.8 | 92.9 | 100.0 | 83.4 |
| LTF [50] | Human | C | 97.4 | 92.8 | 79.0 | 92.4 | 100.0 | 83.8 |
| Transfuser [50] | Human | C&L | 97.7 | 92.8 | 79.2 | 92.8 | 100.0 | 84.0 |
| PARA-Drive [53] | Human | C | 97.9 | 92.4 | 79.3 | 93.0 | 99.8 | 84.0 |
| LAW [17] | Human | C | 96.4 | 95.4 | 81.7 | 88.7 | 99.9 | 84.6 |
| DRAMA [54] | Human | C&L | 98.0 | 93.1 | 80.1 | **94.8** | 100.0 | 85.5 |
| GoalFlow [55] | Human | C&L | 98.3 | 93.8 | 79.8 | 94.3 | 100.0 | 85.7 |
| ARTEMIS [56] | Human | C&L | 98.3 | 95.1 | 81.4 | 94.3 | 100.0 | 87.0 |
| DiffusionDrive [24] | Human | C&L | 98.2 | 96.2 | 82.2 | 94.7 | 100.0 | 88.1 |
| Hydra-MDP [5] | Human & Rule | C&L | 98.3 | 96.0 | 78.7 | 94.6 | 100.0 | 86.5 |
| Hydra-MDP++ [6] | Human & Rule | C | 97.6 | 96.0 | 80.4 | 93.1 | 100.0 | 86.6 |
| WOTE [7] | Human & Rule | C&L | 98.4 | 96.6 | 81.7 | 94.5 | 99.9 | 88.0 |
| **Ours (w/o DPO)** | Human & Rule | C&L | 97.9 | 97.3 | 84.0 | 93.6 | 100.0 | 88.8 |
| **Ours (full)** | Human & Rule | C&L | **98.5** | **98.1** | **84.3** | **94.8** | 99.9 | **90.0** |

Table 2: **Closed-loop results on Bench2Drive.** The best results are in **bold**.

| Method | Efficiency↑ | Comfortness↑ | Success Rate (%)↑ | Driving Score↑ |
|---|---|---|---|---|
| AD-MLP | 48.45 | 22.63 | 0.00 | 18.05 |
| UniAD | 129.21 | 43.58 | 16.36 | 45.81 |
| VAD | 157.94 | **46.01** | 15.00 | 42.35 |
| TCP | 76.54 | 18.08 | 30.00 | 59.90 |
| **Ours** | **166.80** | 26.79 | **30.62** | **62.02** |

Experiments show that both methods significantly improve the safety performance of the policy compared to naive preference construction, with the first method slightly outperforming the second. Therefore, our experiments adopt the first method as the default preference pair construction strategy. Finally, given a constructed preference pair $(a_w, a_l)$, we apply the standard DPO loss:

$$\mathcal{L}_{\text{DPO}} = -\log \sigma \left( \beta \left( \log \frac{\pi_\theta(a_w)}{\pi_{\text{Ref}}(a_w)} - \log \frac{\pi_\theta(a_l)}{\pi_{\text{Ref}}(a_l)} \right) \right) \tag{14}$$

where $\sigma(\cdot)$ is the sigmoid function, $\beta$ is a temperature parameter, and $\pi_{\text{Ref}}$ is the reference policy.

## 5 Experiments

### 5.1 Benchmark

**NAVSIM** NAVSIM [12] benchmark combines real-world sensor data with a non-interactive simulation mechanism, which is built upon OpenScene [57], a reprocessed version of the nuPlan dataset [58]. For each frame, the NAVSIM dataset provides eight high-resolution camera images and fused point cloud data sampled at 2 Hz. The model can take a history of 1.5 seconds as input and is tasked with predicting eight future waypoints over a 4-second horizon. The final standardized training set, Navtrain, contains approximately 103,000 samples, and the test set, Navtest, contains around 12,000 samples. NAVSIM introduces a simulation-based metric called the PDM Score (PDMS), which integrates No At-Fault Collision (NC), Drivable Area Compliance (DAC), Ego Progress (EP), Time-to-Collision (TTC), and Comfort (C) into a single scalar score: PDMS = NC × DAC × (5 × EP + 5 × TTC + 2 × C)/12.

**Bench2Drive**    Bench2Drive [13] is a closed-loop evaluation benchmark for end-to-end autonomous driving. The official training set contains approximately 13,638 short clips, covering 44 categories of interactive scenarios, 23 weather conditions, 12 towns, and a full sensor suite. The evaluation set is organized into 220 short routes that assess various interaction capabilities under different towns and weather. The official closed-loop metrics are Driving Score (DS) and Success Rate (SR), with Driving Efficiency and Comfortness additionally reported; specifically, SR requires the vehicle to reach the destination within the time limit without traffic violations, while DS aggregates route completion and violation penalties into a weighted summary.

## 5.2    Implementation Details

For NAVSIM benchmark, our model is trained on the official NAVSIM [12] training set, and evaluated on the official test set Navtest. For Bench2Drive, our model is trained on base subset and evaluated on the official 220 evaluation routes. We follow the same perception setup and ResNet-34 backbone used in Transfuser for a fair comparison. Specifically, we use a concatenated front-view image of size $1024 \times 256$ formed by three forward-facing cameras as the visual input, fused with a $64 \times 64$ BEV LiDAR feature map. In addition, the model receives a state vector consisting of the current vehicle speed, acceleration, and navigation information. The size of the anchor vocabulary is set to $N = 8192$. We use fixed weights $w_1 = 0.1$ and $w_2 = 1.0$ for unified policy distillation. The number of frequency bands in the positional encoding is set to $L = 10$. The predefined safety threshold $\tau$ is set to 0.3. All experiments are conducted on 6 NVIDIA L20 GPUs, with a batch size of 16 per GPU. We use the AdamW optimizer [59] with a learning rate of $1e-4$. The model is first trained for 30 epochs using unified policy distillation, followed by 10 epochs of fine-tuning with Safety DPO. We sample $K = 1024$ trajectories from the policy distribution for each DPO iteration and set the $\beta = 0.1$. In DPO training, inspired by [11], we introduce an explicit KL regularization term to suppress distributional drift during training. Finally, similar to [35], we continue applying the KL loss from unified policy distillation during the DPO fine-tuning stage as an auxiliary loss.

## 5.3    Comparison with SOTA Methods

We conduct a comprehensive comparison of our method against representative end-to-end baselines on the NAVSIM Navtest split using ResNet-34 as the visual backbone, as shown in Table 1. our method using only unified policy distillation without DPO fine-tuning (Ours w/o DPO) already achieves a PDMS of 88.8, outperforming the current SOTA imitation learning method DiffusionDrive [24] (88.1) and the SOTA score learning method WOTE [7] (88.0). This demonstrates that our unified policy distillation method brings significant performance gains. After applying DPO fine-tuning, our model further improves safety-related metrics: NC increases by 0.6, DAC improves by 0.8, and TTC achieves a gain of 1.2. These results indicate a clear improvement in the safety and reliability of our policy across diverse scenarios. Our method ultimately achieves a new state-of-the-art PDMS of 90.0, outperforming the SOTA imitation learning method [24] by 1.9 and the SOTA score-based method [7] by 2.0. It also surpasses the PDM-closed method [52], which takes the privileged GT Perception. We also conduct closed-loop evaluation on Bench2Drive (Table 2). Our method outperforms representative baselines on Driving Score and Success Rate, demonstrating its effectiveness in closed-loop settings.

## 5.4    Ablation Study

**Ablation on Unified Policy Distillation.**    Table 3 conducts the ablation study to evaluate the effectiveness of the proposed Unified Policy Distillation method. First, compared to the Score Learning method (ID-2), Unified Policy Distillation (ID-1) significantly improves overall policy performance. Furthermore, to assess the importance of combining both imitation and rule-based signals, we conduct additional experiments using single-source supervision. Results show that the Imitation-only method (ID-3) lacks the ability to distinguish high-risk trajectories and performs poorly on critical safety metrics such as DAC, resulting in poor performance. On the other hand, the Rule-only method (ID-4), which ignores human intent, tends to produce aggressive and unreasonable behaviors, leading to suboptimal results in key safety metrics such as NC and TTC.

**Ablation on DPO and Rejected Trajectory Selection.**    Table 4 conducts the ablation study to verify the effectiveness of DPO and the proposed rejected trajectory selection method. Applying

Table 3: **Ablation on Unified Policy Distillation.** Score indicates we regress scores independently for each trajectory anchor. Policy indicates we optimize the policy distribution over all anchors. Sup.: Supervision. The best results are denoted by **bold**.

| ID | Score | Policy | Human Sup. | Rule Sup. | NC↑ | DAC↑ | EP↑ | TTC↑ | PDMS↑ |
|----|-------|--------|------------|-----------|------|------|------|------|-------|
| 1 | ✗ | ✓ | ✓ | ✓ | **97.9** | **97.3** | 84.0 | **93.6** | **88.8** |
| 2 | ✓ | ✗ | ✓ | ✓ | 97.8 | 96.0 | 81.6 | 93.7 | 87.3 |
| 3 | ✗ | ✓ | ✓ | ✗ | 97.6 | 91.2 | 77.5 | 92.9 | 82.5 |
| 4 | ✗ | ✓ | ✗ | ✓ | 95.3 | 97.3 | **85.9** | 88.7 | 87.2 |

Table 4: **Ablation on DPO and Rejected Trajectory Selection.** The best results are denoted by **bold**.

| Method | NC↑ | DAC↑ | EP↑ | TTC↑ | C↑ | PDMS↑ |
|--------|------|------|------|------|------|-------|
| w/o DPO | 97.9 | 97.3 | 84.0 | 93.6 | 100.0 | 88.8 |
| vanilla Selection | 98.4 | 97.5 | 83.5 | 94.6 | 100.0 | 89.3 |
| Distance-Based Selection | 98.1 | **98.2** | **84.3** | 94.2 | 99.9 | 89.7 |
| Imitation-Based Selection | **98.5** | 98.1 | **84.3** | **94.8** | 99.9 | **90.0** |

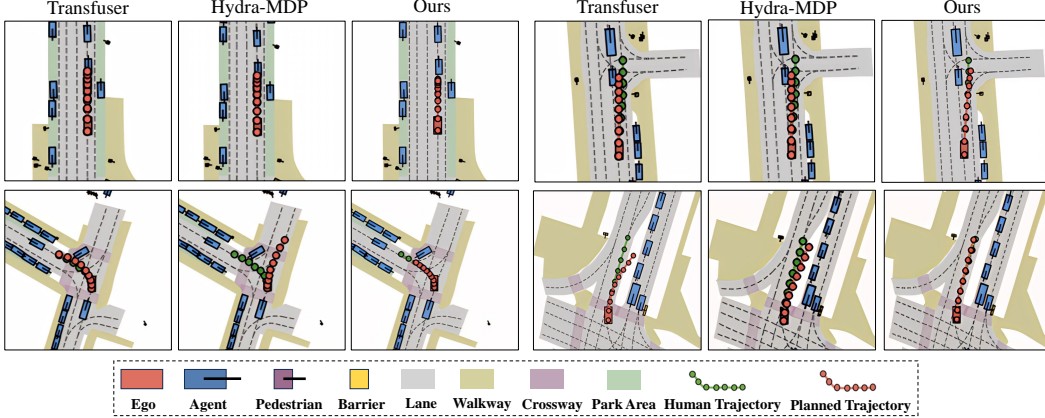

Figure 4: **Qualitative comparison with baselines.** We compare our method against Transfuser [50] and Hydra-MDP [5]. Both the Transfuser and Hydra-MDP suffer from collisions or off-road behaviors in complex road structures, fine-grained turning, and challenging emergency braking tasks. In contrast, our method consistently generates safer trajectories.

the vanilla selection method, which uses the highest and lowest score in $p_{\text{unified}}$ as a preference pair, already improves PDMS by 0.5, indicating that preference-based optimization can effectively suppress low-quality outputs and boost overall performance. Compared to the vanilla selection method, the Distance-Based strategy yields an additional 0.9 improvement in PDMS, and the Imitation-Based strategy improves PDMS by 1.2. This indicates that these strategies further enhance the safety performance by more effectively identifying and suppressing human-like but risky trajectories.

## 5.5 Qualitative Comparison

**Qualitative Comparison with Baselines.** Figure 4 presents a qualitative comparison between our method and two baselines: Transfuser [50], representing imitation learning, and Hydra-MDP [5], representing score-based learning. While Transfuser and Hydra-MDP tend to generate unsafe behaviors such as lane departures or collisions in complex scenarios, our method consistently maintains safe and compliant decisions under diverse conditions.

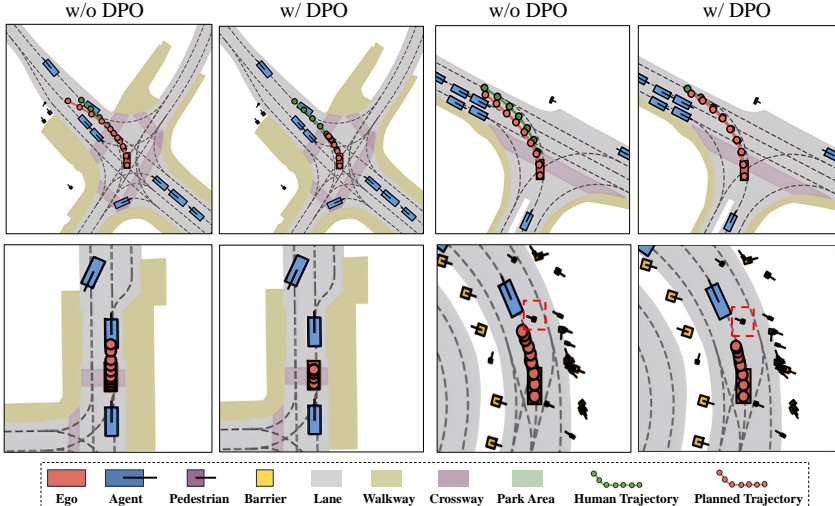

Figure 5: **Qualitative visualization of DPO-enhanced behaviors across diverse scenarios.** The DPO-enhanced policy exhibits more cautious behavior such as early deceleration and smoother cornering, and shows improved responsiveness in challenging scenarios like sudden braking of lead vehicles and pedestrian crossings.

**Qualitative Comparison between Policies with and without DPO.**    To intuitively demonstrate the impact of DPO fine-tuning on policy behavior, we conduct a qualitative analysis of several challenging scenarios as illustrated in Figure 5. The DPO-enhanced policy favors safer and more conservative decisions in complex interaction scenarios, verifying that the DPO fine-tuning improves safety awareness across diverse conditions.

# 6   Limitations and Future Works

Despite the significant progress our method has made in enhancing the safety of end-to-end driving policies, several limitations warrant further study. First, our approach relies on the PDMS metric as the core criterion for safety evaluation. Although PDMS integrates multiple dimensions, including collision avoidance, drivable area compliance, ego progress, time-to-collision, and comfort, it remains a predefined weighted composite metric. As such, it cannot fully capture all potential risk factors in complex driving scenarios. Future work may explore more expressive and flexible trajectory evaluation metrics to build a more comprehensive safety assessment mechanism. Second, our rule-based supervision depends on high-fidelity simulators to provide rule-driven evaluation scores. While effective, the preferences derived from such simulation are inherently limited by the rules' design and the simulator's precision. Moreover, access to such high-fidelity simulators is scarce, constraining the scale and diversity of available data. Therefore, future research should explore preference optimization methods that do not rely on ground-truth perception labels by automatically mining latent preference structures from historical trajectory data. Such efforts could facilitate the development of weakly-supervised or even fully self-supervised safety alignment strategies.

# 7   Conclusion

In this paper, we identify key safety challenges in imitation learning for end-to-end autonomous driving, where models often produce trajectories that appear human-like but are potentially unsafe. In addition, we highlight the limitations of existing score-based methods, which decouple supervision from policy learning and fail to provide direct optimization of the policy distribution. To tackle these challenges, we propose DriveDPO, a Safety DPO Policy Learning framework. By combining unified policy distillation with Safety DPO fine-tuning, DriveDPO enables direct and effective optimization of the policy distribution. Comprehensive experiments and qualitative comparisons on the NAVSIM dataset demonstrate that DriveDPO significantly improves safety and compliance, showing its potential for deployment in safety-critical driving systems.

## Acknowledgements

This work was supported in part by the National Natural Science Foundation of China (No. 62320106010, No. U21B2042).

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
