# OpenReview forum: "DriveDPO: Policy Learning via Safety DPO For End-to-End Autonomous Driving"
_NeurIPS.cc/2025/Conference — NeurIPS 2025 poster_

### Official Review · Reviewer_ndFx · 2025-06-09

**Clarity:** 3
**Significance:** 2
**Originality:** 2
**Rating:** 3
**Confidence:** 5

**Summary:**

The paper introduces a new end-to-end autonomous driving method called DriveDPO. The method has two improvements. The first one is to regress score over all anchors with both human trajectory and rule supervision. The second one is using imitation-based DPO to select safety-aware trajectory pairs for perference learning.

**Questions:**

1. How does DriveDPO perform in highly interactive closed-loop environments with dynamic agents?
2. Could the rejection sample selection process be learned instead of relying on heuristics?
3. Why do more candidate trajectories or DPO fine-tuning epochs lead to poorer performance?

**Ethical Concerns:**

["NO or VERY MINOR ethics concerns only"]

**Final Justification:**

My concern about its closed-loop performance is addressed. But I still think the improvement of DPO is limited and the idea to add rule-based score with imitation-learning score is trival.

**Limitations:**

Yes.

**Quality:**

2

**Strengths And Weaknesses:**

Strengths:

1. This paper has clear motivation.
2. The paper is well-writen and easy to follow.
3. The ablation is extensive, clearly showing the contribution of each part.

Weakness:
1. The method is only evaluated on NAVSIM. The closed-loop performance is not explored.
2. The novelty is limited. The unified policy distillation method to combine the rule-based score and imitation score by weighted sum is trival. The ablation does not compare with the seperate scoring method.
3. The paper only use a one backbone model. Trying more backbone model will be better to demonstrate generality of its method.

---

> ### Author Rebuttal · Authors · 2025-07-30
>
> Thank you for your detailed and constructive reviews. We are glad that you found our paper “has clear motivation” and “is well-writen and easy to follow”. We would like to address the Weaknesses (**W**) and Questions (**Q**) below.
>
> **[W1, Q1] The method is only evaluated on NAVSIM. The closed-loop performance is not explored. How does DriveDPO perform in highly interactive closed-loop environments with dynamic agents?**
>
> Thank you for pointing this out. We conducted closed-loop evaluation experiments on the Bench2Drive dataset [1] (see the table below). Our method outperforms representative baselines in key metrics such as Driving Score, Success Rate, and Mean Multi-Ability Success Rate, demonstrating its effectiveness in closed-loop settings. Notably, the Multi-Ability evaluation directly reflects performance in various highly interactive scenarios. In the particularly challenging Emergency Brake test, DriveDPO surpasses all baseline methods, indicating a strong ability to react to potential risk behaviors. DriveDPO also performs well in interactive tasks such as Overtaking and Give Way, further demonstrating its capability to generate strategies in complex environments with dynamic agents. We will include these closed-loop evaluation results and analysis in the revised version.
>
> | Method | Efficiency | Comfortness | Success Rate(%) | Driving Score |
> | ------ | ---------- | ----------- | --------------- | ------------- |
> | AD-MLP | 48.45      | 22.63       | 0.00            | 18.05         |
> | UniAD  | 129.21     | 43.58       | 16.36           | 45.81         |
> | VAD    | 157.94     | 46.01       | 15.00           | 42.35         |
> | TCP    | 76.54      | 18.08       | 30.00           | 59.90         |
> | Ours   | 166.80     | 26.79       | 30.62           | 62.02         |
>
> | Method | Merging | Overtaking | Emergency Brake | Give Way | Traffic Sign | Mean  |
> | ------ | ------- | ---------- | --------------- | -------- | ------------ | ----- |
> | AD-MLP | 0.00    | 0.00       | 0.00            | 0.00     | 4.35         | 0.87  |
> | UniAD  | 14.10   | 17.78      | 21.67           | 10.00    | 14.21        | 15.55 |
> | VAD    | 8.11    | 24.44      | 18.64           | 20.00    | 19.15        | 18.07 |
> | TCP    | 8.89    | 24.29      | 51.67           | 40.00    | 46.28        | 34.22 |
> | Ours   | 16.28   | 28.95      | 53.06           | 30.00    | 45.00        | 34.66 |
>
>
>
> **[W2.1] The novelty is limited. The unified policy distillation method to combine the rule-based score and imitation score by weighted sum is trival.**
>
> We would like to respectfully clarify that introducing Safety DPO into end-to-end autonomous driving is a non-trivial contribution. First, while prior methods independently predict multiple scores for each anchor to derive a policy implicitly, our approach unifies human-likeness and rule-based safety into a single policy distribution, directly and explicitly supervising policy learning. In addition, to address the human-like but unsafe issue in imitation learning, we propose the Imitation-Based Selection strategy to construct preference pairs that better focus on effective safety preference learning. The effectiveness of our method is consistently demonstrated across both the NAVSIM and Bench2Drive datasets (as in **W1**), as well as under different vision backbones (as in **W3**).
>
>
>
> **[W2.2] The ablation does not compare with the seperate scoring method.**
>
> We have included an ablation study in Table 2 of the main paper (see ID-1 and ID-2), which compares our Unified Policy Distillation with the separate scoring method. The results show that the unified approach significantly outperforms the separate one.
>
>
> **[W3] The paper only use a one backbone model. Trying more backbone model will be better to demonstrate generality of its method.**
>
> Thank you for pointing this out. As shown in the table below, we replaced the vision backbone with more powerful architectures, including V2-99 [2] and ViT-L, and conducted experiments accordingly. The results demonstrate that our method consistently outperforms previous SOTA methods. Moreover, applying DPO fine-tuning further improves performance across all backbone configurations, proving the generality and scalability of our approach. We will include these experimental results and analysis in the revised version.
>
> | Method         | Backbone      | NC   | DAC  | EP   | TTC  | Comf  | PDMS |
> | -------------- | ------------- | ---- | ---- | ---- | ---- | ----- | ---- |
> | GoalFlow       | V2-99         | 98.4 | 98.3 | 85.0 | 94.6 | 100.0 | 90.3 |
> | Hydra-MDP-B    | V2-99         | 98.4 | 97.8 | 86.5 | 93.9 | 100.0 | 90.3 |
> | Hydra-MDP-C    | V2-99 & ViT-L | 98.7 | 98.2 | 86.5 | 95.0 | 100.0 | 91.0 |
> | Hydra-MDP++    | V2-99         | 98.6 | 98.6 | 85.7 | 95.1 | 100.0 | 91.0 |
> | Ours (w/o DPO) | V2-99         | 98.6 | 98.7 | 84.7 | 95.0 | 99.9  | 90.5 |
> | Ours           | V2-99         | 98.9 | 99.1 | 85.2 | 95.9 | 100.0 | 91.4 |
>
> | Method         | Backbone | NC   | DAC  | EP   | TTC  | Comf  | PDMS |
> | -------------- | -------- | ---- | ---- | ---- | ---- | ----- | ---- |
> | Hydra-MDP-A    | ViT-L    | 98.4 | 97.7 | 85.0 | 94.5 | 100.0 | 89.9 |
> | Ours (w/o DPO) | ViT-L    | 98.2 | 98.1 | 84.1 | 94.0 | 100.0 | 89.6 |
> | Ours           | ViT-L    | 98.5 | 99.0 | 84.0 | 95.0 | 100.0 | 90.3 |
>
>
>
> **[Q2] Could the rejection sample selection process be learned instead of relying on heuristics?**
>
> Thank you for the insightful suggestion. We trained a reward model distinguish between *human-like and safe* trajectories versus *human-like but unsafe* ones, and used it for DPO training (see the table below). Compared to relying on rule-based heuristics, using a learned reward model indeed leads to improved performance. This is likely because it can explicitly learn to recognize *human-like but risky* behaviors in ambiguous or complex scenarios that are difficult to capture with hand-crafted rules, thereby improving generalization. We will include this analysis in the revised version. Furthermore, we believe that with access to large-scale, high-quality human preference data in driving scenarios, the learned reward model could be further improved, which is a promising direction for our future work.
>
> | Method                | NC   | DAC  | EP   | TTC  | Comf  | PDMS |
> | --------------------- | ---- | ---- | ---- | ---- | ----- | ---- |
> | w/o DPO               | 97.9 | 97.3 | 84.0 | 93.6 | 100.0 | 88.8 |
> | Rule-based heuristics | 98.4 | 97.5 | 83.5 | 94.6 | 100.0 | 89.3 |
> | Learned reward model  | 98.2 | 98.2 | 83.7 | 94.3 | 100.0 | 89.6 |
>
>
>
> **[Q3] Why do more candidate trajectories or DPO fine-tuning epochs lead to poorer performance?**
>
> When the number of candidate trajectories K is overly increased, the selected rejected samples often have very low probabilities under the current policy distribution. As a result, they may contribute little to the policy optimization, ultimately degrading performance. As for the drop in performance after 20 epochs of DPO fine-tuning, this is potentially due to overfitting on the training set. We empirically set the number of candidate trajectories to 1024 and fine-tuned the policy with DPO for 10 epochs, which yielded the best performance in our experiments.
>
>
>
> [1] Jia, Xiaosong, et al. "Bench2drive: Towards multi-ability benchmarking of closed-loop end-to-end autonomous driving." NeurIPS 2024.
>
> [2] Lee, Youngwan, and Jongyoul Park. "Centermask: Real-time anchor-free instance segmentation." CVPR 2020.

---

> > ### Comment · Reviewer_ndFx · 2025-08-05
> > **Response  to rebuttal**
> >
> > My concern about its closed-loop performance is addressed.
> > But I still think the improvement of DPO is limited (compared with Hydra-mdp). The idea to add rule-based score with imitation-learning score is trival.
> > And the seperate scoring method in ablation study in Table 2 should be make more clear.
> > I have raised my score.

---

> ### Author Response · Authors · 2025-08-05
> **Response to Reviewer ndFx's Comment**
>
> We sincerely appreciate your positive follow-up and for raising your score, as well as for giving us the opportunity to further improve our work through your valuable comments. Below, we provide further clarifications and responses to your remaining concerns, and hope these can help address them.
>
> **1. The improvement of DPO is limited (compared with Hydra-mdp).**
>
> Under the same ResNet-34 backbone, our method achieves a PDMS improvement of +3.4 over Hydra-MDP. Considering that previous works reported performance gains over the SOTA of +1.6 in [1], +1.8 in [2], and +1.3 in [3], we believe this constitutes a significant improvement. In addition, applying Safety DPO further yields a +1.2 PDMS increase, which demonstrates its effectiveness. When using other backbones, our method still outperforms Hydra-MDP in PDMS by +0.4, with larger relative gains in safety-related metrics (+1.3 DAC with ViT-L, +0.8 TTC with V2-99). We would also like to note that these experiments with alternative backbones were conducted within one week during the rebuttal period, and we believe that with more thorough hyperparameter tuning, further improvements are achievable. We will update these results in the revised version.
>
> **2. The idea to add rule-based score with imitation-learning score is trivial.**
>
> We would like to respectfully clarify that our main contribution lies in introducing Safety DPO into end-to-end autonomous driving. Our key finding is that pure imitation learning often favors human-like but unsafe trajectories. Safety DPO addresses this by reformulating training as trajectory-level preference alignment, explicitly preferring trajectories that are both human-like and safe. We would also like to clarify that the unified policy distillation, which combines imitation similarity with rule-based safety scores, simultaneously considers human preference and rule-based constraints, and merges them into a unified probability distribution over trajectories, enabling direct and coherent policy supervision.
>
> **3. The seperate scoring method in ablation study in Table 2 should be make more clear.**
>
> Thank you for pointing this out. In the revised version, we will explicitly state in the caption of Table 2 that ID-2 corresponds to the separate scoring method. This will ensure that readers can clearly understand the setting and fairly interpret the ablation results.
>
>
>
> [1] Liao, Bencheng, et al. "Diffusiondrive: Truncated diffusion model for end-to-end autonomous driving." CVPR. 2025.
>
> [2] Li, Yingyan, et al. "End-to-end driving with online trajectory evaluation via bev world model." ICCV. 2025.
>
> [3] Zheng, Yupeng, et al. "World4Drive: End-to-End Autonomous Driving via Intention-aware Physical Latent World Model." ICCV. 2025.

---

### Official Review · Reviewer_o9SK · 2025-06-17

**Clarity:** 3
**Significance:** 3
**Originality:** 3
**Rating:** 5
**Confidence:** 4

**Summary:**

This paper proposes to fine-tune trajectory scoring based end-to-end driving models with direct preference optimization by selecting positive and negative trajectories. The goal is to filter out trajectories that are close to a human reference trajectory in euclidean distance but unsafe.
As positive trajectory the highest scoring trajectory is used. The negative trajectory either uses a trajectory with low distance to ground truth or low distance to the best trajectory, that still has low PDMS.
DPO is used to finetune a basemodel which predicts a distribution over anchor trajectories.
The method is evaluated and ablated on the NavSim benchmark.

**Questions:**

What is the inference speed of the method, and how long did training with the 6 L20 GPUs take?
The introduction claims that indirect optimization leads to suboptimal driving performance. It is not clear to me why this should be the case.

**Ethical Concerns:**

["NO or VERY MINOR ethics concerns only"]

**Final Justification:**

My concerns have been adressed. I have updated my rating to accept.

**Limitations:**

yes

**Paper Formatting Concerns:**

-

**Quality:**

3

**Strengths And Weaknesses:**

The paper follows the standard formula of proposing a method, comparing it to baselines, and ablating its components.
It expands on the recent trend in end-to-end driving of optimizing the target metric (PDMS) via scoring a set of anchor trajectories. The paper proposes to use the DPO method from the RLHF community for this optimization. A novel way to first learn a distribution over trajectories and then select positive and negative samples for DPO fine-tuning is proposed. Since this is an active research direction, the work should be interesting for many researchers in the field.

Weaknesses:

The abstract claims that DriveDPO achieves a state-of-the-art PDMS of 90.0. This is incorrect because the prior SOTA is Hydra-MDP-C [3] with 91.0 PDMS.

Figure 1 a) cites Uni-AD and VAD. Both of these papers do not use Anchor Vocabularies as depicted in the figure.
This idea was developed by VADv2 (which is also cited). It would make more sense to only cite VADv2 here.

NAVSIM [1] should be cited when discussing the downside of the L2 learning objective, as it has also made these points.

L. 53: This sentence could be formulated clearer such that it is obvious to the reader that only methods using ResNet-34 as backbone are outperformed (and not other methods using bigger vision backbones).

Claim (1) that this paper is the first to point out the problem of imitation learning  (L1/L2 loss) seems to be not correct. For example, this problem was already discussed in NAVSIM Figure 1 [1].

L70: The introduction to end-to-end autonomous driving is somewhat inaccurate. It is claimed that these methods predict trajectories from raw sensor data. Predicting trajectories is recently quite popular in the end-to-end driving literature but does not characterize these techniques. Many methods [2, 4, 5, 6, 7, 8, 9, 10, 11, 12] also directly predict the control of the car, in particular earlier works from before 2021.

Section 2.2 is missing earlier work on fine-tuning / training end-to-end architectures with reinforcement learning, such as [9, 10, 11, 12].

L. 220 / 221: The paper incorrectly states that WOTE is the SOTA trajectory scoring method with 88.0 PDMS. This is incorrect because Hydra-MDP-C (CVPR 2024 Workshop) [3] is the prior work SOTA with 91.0 PDMS.
The second claim here is more nuanced. DiffusionDrive is claimed to be the SOTA imitation learning method.
DiffusionDrive still technically scores trajectories, although with probabilities and not predicted metrics.
So there is an argument to be made that it can be categorized differently than methods like Hydra that score based on metrics, but the wording needs to be precise.

L. 225: See above; this claim is not entirely correct.

Table 1: Several methods are misrepresented in the Table 1 comparison:

    • Hydra-MDP: The performance of the random ablation Hydra-MDP-V8192-W-EP is quoted as Hydra-MDP (86.5 PDMS). Instead, the best Hydra-MDP version, Hydra-MDP-C (91 PDMS), should be cited.
    • WoTE: WoTE is quoted as achieving 88.0 PDMS, but the paper reports 87.1 PDMS.
    • Hydra-MDP++: The best version of Hydra-MDP++ achieves 91.0 PDMS, but the weaker result with a smaller backbone and 86.6 PDMS is cited.
    • GoalFlow: GoalFlow is quoted to achieve 85.7 PDMS (perhaps the random ablation Dim 256 Backbone resnet34 is referenced here), but the reported best performance of the method is 90.3 PDMS.

If the authors want to compare to methods using the same vision backbone (ResNet34) they can make the vision backbone a column and compare within the group (but also report the best performance of each method in another group / table).


So to summarize, I think the paper proposes an interesting idea that is of value to the community.
However, I am concerned that the experimental evaluation may mislead the reader by overselling the achieved performance, via misrepresentation of baselines. I do not think it is necessary to make a state-of-the-art claim for publication, in particular because the performance difference is small, but I do think it is important to properly represent baselines and make correct and precise claims.
I think the paper should be improved by making the claims more specific (e.g., outperforming methods that use the ResNet34 backbone), properly reporting the best performance of baselines, and more rigorously citing prior work.
I would consider raising my score if these concerns are addressed.

If the authors want to make a claim around advancing the state-of-the-art on NavSim I suggest reporting the performance of the method with competitive vision backbones like VIT-L or V2-99 instead of / in addition to ResNet34.

References:

[1] Daniel Dauner, Marcel Hallgarten, Tianyu Li, Xinshuo Weng, Zhiyu Huang, Zetong Yang, Hongyang Li, Igor Gilitschenski, Boris Ivanovic, Marco Pavone, Andreas Geiger, Kashyap Chitta: “NAVSIM: Data-Driven Non-Reactive Autonomous Vehicle Simulation and Benchmarking”, NeurIPS 2024

[2] Penghao Wu, Xiaosong Jia, Li Chen, Junchi Yan, Hongyang Li, Yu Qiao: “Trajectory-guided Control Prediction for End-to-end Autonomous Driving: A Simple yet Strong Baseline”, NeurIPS 2022

[3] Zhenxin Li, Kailin Li, Shihao Wang, Shiyi Lan, Zhiding Yu, Yishen Ji, Zhiqi Li, Ziyue Zhu, Jan Kautz, Zuxuan Wu, Yu-Gang Jiang, Jose M. Alvarez: “Hydra-MDP: End-to-end Multimodal Planning with Multi-target Hydra-Distillation”,  CVPR 2024 Workshop

[4] Dean A. Pomerleau: “ALVINN: AN AUTONOMOUS LAND VEHICLE IN A NEURAL NETWORK”, NeurIPS 1988

[5] Mariusz Bojarski, Davide Del Testa, Daniel Dworakowski, Bernhard Firner, Beat Flepp, Prasoon Goyal, Lawrence D. Jackel, Mathew Monfort, Urs Muller, Jiakai Zhang, Xin Zhang, Jake Zhao, Karol Zieba: “End to End Learning for Self-Driving Cars”,  ArXiv 2016

[6] Felipe Codevilla, Matthias Müller, Antonio López, Vladlen Koltun, Alexey Dosovitskiy: “End-to-end Driving via Conditional Imitation Learning” ICRA 2018

[7] Felipe Codevilla, Eder Santana, Antonio M. López, Adrien Gaidon: “Exploring the Limitations of Behavior Cloning for Autonomous Driving”, ICCV 2019

[8]  Dian Chen, Brady Zhou, Vladlen Koltun, Philipp Krähenbühl: “Learning by Cheating.”, CoRL 2019

[9] Alex Kendall, Jeffrey Hawke, David Janz, Przemyslaw Mazur, Daniele Reda, John-Mark Allen, Vinh-Dieu Lam, Alex Bewley, Amar Shah: “Learning to Drive in a Day.”, ICRA 2019

[10] Marin Toromanoff, Emilie Wirbel, Fabien Moutarde: “End-to-End Model-Free Reinforcement Learning for Urban Driving using Implicit Affordances”, CVPR 2020

[11] Xiaodan Liang, Tairui Wang, Luona Yang, Eric Xing: “CIRL: Controllable Imitative Reinforcement Learning for Vision-based Self-driving”, 2018

[12] Raphael Chekroun, Marin Toromanoff, Sascha Hornauer, Fabien Moutarde: “GRI: General Reinforced Imitation and its Application to Vision-Based Autonomous Driving”, Robotics 2023

---

> ### Author Rebuttal · Authors · 2025-07-30
>
> Thank you for your detailed and constructive reviews. We are glad that you found that “the paper proposes an interesting idea that is of value to the community” and “the work should be interesting for many researchers in the field”. We would like to address the Weaknesses (**W**) and Questions (**Q**) below.
>
> **[W1] Reporting the performance of the method with competitive vision backbones in addition to ResNet34.**
>
> We sincerely thank you for pointing out the issues regarding accurate citation of baseline performance and fair comparisons. We conducted additional experiments using stronger vision backbones V2-99 and ViT-L (see table below). Our method still outperforms existing SOTA approaches, including Hydra-MDP-A [1], Hydra-MDP-C [1], and Hydra-MDP++ [2]. In the revised version, we will revise our claims to clarify that DriveDPO achieves state-of-the-art performance when utilizing the same vision backbone, and we will explicitly categorize methods by backbone to ensure a fair comparison. We will also correct and include proper citations, as well as the best-reported performance of Hydra-MDP-C [1], Hydra-MDP++ [2], and GoalFlow [3].
>
> | Method         | Backbone      | NC   | DAC  | EP   | TTC  | Comf  | PDMS |
> | -------------- | ------------- | ---- | ---- | ---- | ---- | ----- | ---- |
> | GoalFlow       | V2-99         | 98.4 | 98.3 | 85.0 | 94.6 | 100.0 | 90.3 |
> | Hydra-MDP-B    | V2-99         | 98.4 | 97.8 | 86.5 | 93.9 | 100.0 | 90.3 |
> | Hydra-MDP-C    | V2-99 & ViT-L | 98.7 | 98.2 | 86.5 | 95.0 | 100.0 | 91.0 |
> | Hydra-MDP++    | V2-99         | 98.6 | 98.6 | 85.7 | 95.1 | 100.0 | 91.0 |
> | Ours (w/o DPO) | V2-99         | 98.6 | 98.7 | 84.7 | 95.0 | 99.9  | 90.5 |
> | Ours           | V2-99         | 98.9 | 99.1 | 85.2 | 95.9 | 100.0 | 91.4 |
>
> | Method         | Backbone | NC   | DAC  | EP   | TTC  | Comf  | PDMS |
> | -------------- | -------- | ---- | ---- | ---- | ---- | ----- | ---- |
> | Hydra-MDP-A    | ViT-L    | 98.4 | 97.7 | 85.0 | 94.5 | 100.0 | 89.9 |
> | Ours (w/o DPO) | ViT-L    | 98.2 | 98.1 | 84.1 | 94.0 | 100.0 | 89.6 |
> | Ours           | ViT-L    | 98.5 | 99.0 | 84.0 | 95.0 | 100.0 | 90.3 |
>
>
>
> **[W2] Figure 1 a) cites Uni-AD and VAD. It would make more sense to only cite VADv2 here.**
>
> Thank you for pointing this out. We will revise the caption to cite only VADv2, reflecting the correct origin of the idea.
>
>
>
> **[W3] NAVSIM should be cited when discussing the downside of the L2 learning objective. Claim that this paper is the first to point out the problem of imitation learning (L1/L2 loss) seems to be not correct. This problem was already discussed in NAVSIM Figure 1.**
>
> We would like to clarify that NAVSIM [4] discusses the limitations of L1/L2-based evaluation metrics such as ADE, but does not discuss the issue of using L1/L2 objectives as training losses. That said, we acknowledge that the insight raised in NAVSIM aligns with the motivation of our work. Our contribution lies in being the first to extend this perspective to the training objective and demonstrate clear empirical benefits from doing so. We will include a citation and discussion of NAVSIM in the revised version.
>
>
>
> **[W4] L53: This sentence could be formulated clearer such that it is obvious to the reader that only methods using ResNet-34 as backbone are outperformed.**
>
> Thank you for catching this ambiguity. We will revise the sentence to: “We outperform all prior methods that use ResNet-34 as the vision backbone.”
>
>
>
> **[W5] L70: The introduction to end-to-end autonomous driving is somewhat inaccurate. Many methods also directly predict the control of the car, in particular earlier works from before 2021.**
>
> We appreciate this correction. We will revise this to: “End-to-end autonomous driving typically maps raw sensor inputs to driving actions, either in the form of trajectories or low-level control commands [5–14].”
>
>
>
> **[W6] Section 2.2 is missing earlier work on fine-tuning / training end-to-end architectures with reinforcement learning.**
>
> In the revised version, we will add a paragraph in Section 2.2 to briefly review the earlier work on training end-to-end architectures with reinforcement learning: “Kendall et al. [11] demonstrated an on-vehicle deep RL system for lane following using monocular input and distance-based reward. Toromanoff et al. [12] introduced implicit affordances to enable model-free RL in urban settings with traffic light and obstacle handling. Liang et al. [13] proposed CIRL, combining goal-conditioned RL and human demonstration to improve success rates in CARLA. Chekroun et al. [14] developed General Reinforced Imitation, which integrates expert data into off-policy RL for stable vision-based urban driving.”
>
>
>
> **[W7] L. 220 / 221: The paper incorrectly states that WOTE is the SOTA trajectory scoring method with 88.0 PDMS. This is incorrect because Hydra-MDP-C is the prior work SOTA with 91.0 PDMS. DiffusionDrive is claimed to be the SOTA imitation learning method.**
>
> We appreciate your pointing out the inaccuracies regarding prior work. To avoid confusion, we will restrict the comparison scope to methods using ResNet-34 and revise the original description to: “ Using ResNet-34 as the visual backbone, our method with only unified policy distillation achieves a PDMS of 88.8, outperforming the diffusion-based method DiffusionDrive, which scores trajectories through imitation probabilities, and the metric-based scoring method WOTE. ”
>
>
>
> **[W8] L. 225: See above; this claim is not entirely correct.**
>
> We will clarify that our method achieves a new SOTA among methods using the ResNet-34 backbone by modifying the sentence to: “Our method ultimately achieves a PDMS of 90.0, establishing a new state-of-the-art among methods using ResNet-34 as the visual backbone.” As discussed in **W1**, we will further include comparisons and analysis with SOTA methods like Hydra-MDP-C using V2-99 and ViT-L as backbones in the revised version.
>
>
>
> **[W9] Table 1: Several methods are misrepresented in the Table 1 comparison**
>
> To ensure fair comparison, we will group methods using the same visual backbone in Table 1 and clearly distinguish between the performance of methods based on different backbone architectures. We will include the best reported performances and citations for Hydra-MDP-C [1], Hydra-MDP++ [2], and GoalFlow [3] in the revised version. For WoTE [15], we will update its performance to 88.3 as reported in its most recent arXiv version.
>
>
>
> **[Q1] What is the inference speed of the method, and how long did training with the 6 L20 GPUs take?**
>
> The table below presents the inference latency of different methods (some results are cited from Hydra-MDP++ [2]). Our method achieves faster inference speed under both ResNet-34 and V2-99 backbones. Additionally, training on 6 × L20 GPUs takes approximately 10 minutes per epoch, totaling around 7 hours for 40 epochs.
>
> | Method      | Backbone | Test GPU    | Latency (ms) |
> | ----------- | -------- | ----------- | ------------ |
> | Transfuser  | Resnet34 | NVIDIA V100 | 221.2        |
> | UniAD       | Resnet34 | NVIDIA A100 | 555.6        |
> | Hydra-MDP++ | Resnet34 | NVIDIA V100 | 206.2        |
> | Ours        | Resnet34 | NVIDIA L20  | 137.8        |
> | Hydra-MDP++ | V2-99    | NVIDIA V100 | 271.0        |
> | Ours        | V2-99    | NVIDIA L20  | 240.8        |
>
>
>
> **[Q2] The introduction claims that indirect optimization leads to suboptimal driving performance. It is not clear to me why this should be the case.**
>
> Previous score-based methods regress multiple scores for each anchor using several metric heads in a multi-task learning manner. However, the objectives of different metrics may conflict with each other, leading to inconsistent or even conflicting gradient directions, which can adversely affect overall performance.
>
>
>
> [1] Li, Zhenxin, et al. "Hydra-mdp: End-to-end multimodal planning with multi-target hydra-distillation." arXiv:2406.06978.
>
> [2] Li, Kailin, et al. "Hydra-mdp++: Advancing end-to-end driving via expert-guided hydra-distillation." arXiv:2503.12820.
>
> [3] Xing, Zebin, et al. "Goalflow: Goal-driven flow matching for multimodal trajectories generation in end-to-end autonomous driving." CVPR 2025.
>
> [4] Dauner, Daniel, et al. "Navsim: Data-driven non-reactive autonomous vehicle simulation and benchmarking." NeurIPS 2024.
>
> [5] Penghao Wu, et al. "Trajectory-guided Control Prediction for End-to-end Autonomous Driving: A Simple yet Strong Baseline" NeurIPS 2022.
>
> [6] Dean A. Pomerleau, et al. “ALVINN: AN AUTONOMOUS LAND VEHICLE IN A NEURAL NETWORK” NeurIPS 1988.
>
> [7] Bojarski, Mariusz, et al. "End to end learning for self-driving cars." arXiv:1604.07316.
>
> [8] Felipe Codevilla, et al. "End-to-end Driving via Conditional Imitation Learning" ICRA 2018
>
> [9] Felipe Codevilla, et al. "Exploring the Limitations of Behavior Cloning for Autonomous Driving" ICCV 2019
>
> [10] Dian Chen, et al. "Learning by Cheating." CoRL 2019
>
> [11] Alex Kendall, et al. "Learning to Drive in a Day." ICRA 2019
>
> [12] Marin Toromanoff, et al. "End-to-End Model-Free Reinforcement Learning for Urban Driving using Implicit Affordances" CVPR 2020
>
> [13] Xiaodan Liang, et al. "CIRL: Controllable Imitative Reinforcement Learning for Vision-based Self-driving" 2018
>
> [14] Raphael Chekroun, et al. "GRI: General Reinforced Imitation and its Application to Vision-Based Autonomous Driving" Robotics 2023
>
> [15] Li, Yingyan, et al. "End-to-end driving with online trajectory evaluation via bev world model." arXiv:2504.01941.

---

> > ### Comment · Reviewer_o9SK · 2025-08-03
> >
> > Thank you for answering my questions.
> > I have an additional question regarding your response.
> > The inference times seem a bit strange.
> > I would have expected TransFuser with a ResNet34 backbone to be real-time < 50 ms.
> > It should also have similar or lower inference than your method, although I guess the difference here comes from the newer GPU (V100, L20)?
> > What input resolutions for Lidar and Camera images are being used here?
> >
> > I guess these are the numbers copied from Hydra++.
> > Can you measure the publicly available methods on the same GPU for a fair comparison?

---

> > > ### Author Response · Authors · 2025-08-04
> > > **Response to Reviewer o9SK's Comment**
> > >
> > > We sincerely appreciate your response and follow-up question. Same as TransFuser, we use an image resolution of $256 \times 1024$ and a LiDAR resolution of $256 \times 256$. Regarding the latency, the higher values reported earlier were due to using a batch size of 4 during measurement (our initial testing setting). In the updated experiments, we standardized the settings to batch size = 1 and GPU = NVIDIA L20 for all methods.
> > >
> > > The table below reports the measured latency of TransFuser, Hydra-MDP, and our method under these standardized settings. Since Hydra-MDP is not publicly available, we measured its latency using our own reimplementation. We observe that our method is slightly slower than TransFuser due to a larger number of tokens in the Transformer decoder, while Hydra-MDP is slightly slower than ours because it predicts multiple score heads. We hope this clarifies the issue, and sincerely look forward to your reply.
> > >
> > > | Method     | Backbone | Latency (ms) |
> > > | ---------- | -------- | ------------ |
> > > | Transfuser | Resnet34 | 30.6         |
> > > | Hydra-MDP  | Resnet34 | 39.1         |
> > > | Ours       | Resnet34 | 37.9         |
> > > | Hydra-MDP  | V2-99    | 51.2         |
> > > | Ours       | V2-99    | 49.8         |

---

> ### Comment · Reviewer_o9SK · 2025-08-04
>
> My concerns have been adressed. I have updated my rating to accept.

---

> > ### Author Response · Authors · 2025-08-04
> >
> > Thank you for updating your rating to accept. We truly appreciate your recognition of our work and the opportunity to further improve it through your valuable comments.

---

### Official Review · Reviewer_2wBa · 2025-07-03

**Clarity:** 4
**Significance:** 4
**Originality:** 3
**Rating:** 5
**Confidence:** 3

**Summary:**

DriveDPO introduces a two-stage policy learning framework for end-to-end autonomous driving by using both imitation and rule-based safety signals and refining the policy using Safety DPO.

It achieves state-of-the-art performance on the NAVSIM benchmark, significantly improving safety-related metrics.

**Questions:**

- Are the constructed preferences always meaningful and safe? There could be some analysis on this.
- Can DriveDPO function effectively without a high-fidelity simulator?

**Ethical Concerns:**

["NO or VERY MINOR ethics concerns only"]

**Final Justification:**

I believe the authors have addressed most of my and other reviewers' concerns. I also think the simplicity of the work is a strength of this paper. So I have raised my score.

**Limitations:**

- Statistical Robustness (computational resources limitations pointed in the paper)
- Fixed Perception Backbone: The policy model uses a fixed ResNet/Transfuser backbone

**Quality:**

2

**Strengths And Weaknesses:**

Strengths of the Approach
1. Safety-Aligned Policy Learning: Combines imitation learning and rule-based safety via a unified distribution
2. Direct Optimization of Policy Distribution: Unlike score-based approaches, DriveDPO optimizes the actual policy. Introduces Safety DPO
3. Achieves a new state-of-the-art.


Weaknesses of the Approach
1. Dependency on Rule-Based Scores
    - Requires a high-fidelity simulator (NAVSIM) to generate rule-based metrics (PDMS), which may not generalize well to all driving environments or be available in real-world settings. So this might not translate to training in real-world.
    - Preference pairs are constructed using rule thresholds and heuristics. This may not reflect the full nuance of real-world human preferences or corner cases.
2. Anchor Vocabulary Limits Output Flexibility: The discretization of trajectory space into anchors, while efficient, may limit expressiveness or fail to capture rare but valid driving maneuvers.
3. No Online Learning or Interaction: The system is trained offline and evaluated in simulation; there’s no feedback loop with deployment environments.

Weaknesses / Gaps in the Work
1. No Exploration of Failure Cases: The paper doesn't systematically analyze where DriveDPO fails (e.g., rare edge cases, adversarial driving).
2. No Ablation on DPO Parameters
3. While many RLHF methods are cited, DriveDPO is not directly compared with methods using reward modeling + PPO (e.g., GenDrive, TrajHF) on the same benchmark.
4. Simulation-only Evaluation: The policy is not validated in a closed-loop interactive simulator setting

---

> ### Author Rebuttal · Authors · 2025-07-30
>
> Thank you for your detailed and constructive reviews. We are glad that you found that our method “significantly improving safety-related metrics”. We would like to address the Weaknesses (**W**), Questions (**Q**), and Limitations (**L**) below.
>
> **[W1.1, Q2] Requires a high-fidelity simulator to generate rule-based metrics. Can DriveDPO function effectively without a high-fidelity simulator?**
>
> We understand your concerns. Actually, computing rule-based scores only relies on a few elements provided by most driving datasets: the ego vehicle’s states and trajectories, lane geometry, and the positions of dynamic objects. Rule-based scores can be derived from geometric relationships between trajectories and labeled information, so there is no dependency on a high-fidelity simulator. We also computed rule-based scores on the Bench2Drive dataset and successfully used them for training (see results in **W3**), demonstrating the potential applicability of our method to real-world training scenarios.
>
> **[W1.2] Preference pairs are constructed using rule thresholds and heuristics. This may not reflect the full nuance of real-world human preferences or corner cases.**
>
> Thank you for the insightful suggestion. As shown in the table below, we employed a learned reward model to construct preference pairs for DPO training, which proved to be effective. This may be because it can learn human preferences in ambiguous or complex scenarios that cannot be well captured by rules, thus providing better generalization. We will include this experimental analysis in the revised version. We also believe that with access to large-scale and high-quality human preference datasets, which are currently lacking in the driving community, the model performance could be further improved. This will be an important direction we plan to explore in future work.
>
> | Method| NC| DAC  | EP| TTC  | Comf  | PDMS |
> | -------------------- | ---- | ---- | ---- | ---- | ----- | ---- |
> | w/o DPO| 97.9 | 97.3 | 84.0 | 93.6 | 100.0 | 88.8 |
> | Rule-based reward| 98.4 | 97.5 | 83.5 | 94.6 | 100.0 | 89.3 |
> | Learned reward model | 98.2 | 98.2 | 83.7 | 94.3 | 100.0 | 89.6 |
>
> **[W2] Anchor Vocabulary Limits Output Flexibility.**
>
> We understand your concerns. In fact, anchor-based planners can flexibly compose complex driving behaviors by selecting different anchor trajectories at different time steps, thus enabling coverage of rare but valid maneuvers. We also conducted Multi-Ability evaluations on the Bench2Drive dataset (see **W3**), including tasks like Merging, Overtaking, and Give Way. The competitive results indicate that the anchor-based strategy does not limit the model’s ability to express complex driving behaviors.
>
> **[W3] The policy is not validated in a closed-loop interactive simulator setting. The system is trained offline and evaluated in simulation.**
>
> We conducted a closed-loop evaluation on the Bench2Drive dataset [1] (shown in the table below). The results show that our method outperforms representative baselines in key metrics, including Driving Score, Success Rate, and Mean Multi-Ability Success Rate. In particular, our method achieves the best performance in the Emergency Brake scenario, indicating the policy’s sensitivity to potentially risky behaviors. We will include the closed-loop results and analysis in the revised version.
>
> | Method | Efficiency | Comfortness | Success Rate(%) | Driving Score |
> | ------ | ---------- | ----------- | --------------- | ------------- |
> | AD-MLP | 48.45| 22.63| 0.00| 18.05|
> | UniAD  | 129.21| 43.58| 16.36 | 45.81|
> | VAD    | 157.94| 46.01| 15.00 | 42.35|
> | TCP    | 76.54| 18.08 | 30.00  | 59.90|
> | Ours   | 166.80| 26.79| 30.62 | 62.02|
>
> | Method | Merging | Overtaking | Emergency Brake | Give Way | Traffic Sign | Mean  |
> | ------ | ------- | ---------- | --------------- | -------- | ------------ | ----- |
> | AD-MLP | 0.00| 0.00| 0.00| 0.00| 4.35| 0.87  |
> | UniAD  | 14.10| 17.78| 21.67| 10.00| 14.21| 15.55 |
> | VAD    | 8.11| 24.44| 18.64| 20.00| 19.15| 18.07 |
> | TCP    | 8.89| 24.29| 51.67| 40.00| 46.28| 34.22 |
> | Ours   | 16.28| 28.95| 53.06| 30.00| 45.00| 34.66 |
>
> **[W4] No Exploration of Failure Cases.**
>
> We found that DriveDPO sometimes adopts more conservative driving strategies, which may reduce forward efficiency in specific scenarios. As shown in the table below, we computed the Ego Progress metric for non-collision cases and observed a decrease after DPO fine-tuning. This suggests that the policy makes a more conservative trade-off between safety and efficiency, which could result in suboptimal performance in interactions that require faster speeds. We will include the failure case analysis in the revised version.
>
> | Method| Ego Progress in Succeeded Case |
> | --------------- | ------------------------------ |
> | Baseline| 80.73|
> | w/ DPO Finetune | 80.19|
>
> **[W5] No Ablation on DPO Parameters.**
>
> The ablation results are provided in Tables 1–3 of the supplementary material. We conducted systematic ablation studies and analysis on key DPO parameters, including the number of candidate trajectories $K$, the number of DPO fine-tuning epochs, and the starting epoch of DPO.
>
> **[W6] DriveDPO is not directly compared with methods using reward modeling + PPO (e.g., GenDrive, TrajHF) on the same benchmark.**
>
> We thank you for the suggestion. However, GenDrive is not an end-to-end driving method and thus cannot be directly compared on the same benchmark. In addition, TrajHF relies on a private preference dataset, making it difficult to reproduce and compare fairly. To enable a fair comparison, we implemented a variant using reward modeling and PPO (see table below). While this approach shows improvements over the baseline, it still underperforms compared to our method. This may be due to the instability of PPO training, whereas our DriveDPO leverages high-quality preferences with more stable safety alignment. We also noticed a concurrent work, ReCogDrive [2], which adopts GRPO and is evaluated on the NAVSIM benchmark. Despite not using VLM as ReCogDrive, our proposed DriveDPO still outperforms ReCogDrive, indicating the advantages of our method.
>
> | Method| NC | DAC| EP| TTC|Comf| PDMS |
> | ----------------------- | ---- | ---- | ---- | ---- | ----- | ---- |
> | Baseline| 97.9 | 97.3 | 84.0 | 93.6 | 100.0 | 88.8 |
> | Reward modeling and PPO | 97.9 | 98.1 | 84.1 | 94.0 | 100.0 | 89.4 |
> | ReCogDrive (GRPO) | 98.2 | 97.8 | 83.5 | 95.2 | 99.8 | 89.6 |
> | Ours| 98.5 | 98.1 | 84.3 | 94.8 | 99.9  | 90.0 |
>
>
> **[Q1] Are the constructed preferences always meaningful and safe?**
>
> We analyzed the safety and human-likeness of the constructed preference pairs. The table below reports the average safety score and Human Trajectory Distance for all sampled anchor trajectories, as well as for the chosen and rejected ones. We observe that the chosen trajectories have average safety scores close to 100, whereas the rejected ones have scores near 0. Both are also close to the human trajectory in terms of distance. This indicates that the constructed preference pairs effectively distinguish between *human-like but unsafe* and *human-like and safe* behaviors, demonstrating their reliability and validity.
>
> | Trajectory Set| Safety Score | Human Trajectory Distance |
> | ------------------------------- | ------------ | ------------------------- |
> | All sampled anchor trajectories | 26.92| 24.87|
> | Chosen trajectories| 93.34| 0.77|
> | Rejected trajectories| 8.68| 1.99|
>
> **[L1] Statistical Robustness**
>
> We conducted five independent training runs of our model and report the mean and standard deviation of key metrics in the table below. The results show low variance across runs, indicating that our method exhibits strong statistical robustness.
>
> | Method   | NC| DAC          | EP           | TTC          | Comf         | PDMS         |
> | -------- | ------------ | ------------ | ------------ | ------------ | ------------ | ------------ |
> | Baseline | 97.9         | 97.3         | 84.0         | 93.6         | 100.0        | 88.8         |
> | Ours     | 98.27 ± 0.17 | 98.13 ± 0.05 | 84.17 ± 0.34 | 94.43 ± 0.26 | 99.97 ± 0.05 | 89.77 ± 0.17 |
>
> **[L2] Fixed Perception Backbone**
>
> As shown in the table below, we replaced the vision backbone of the policy model with more powerful architectures, including V2-99 [3] and ViT-L. The results show that our method consistently outperforms previous SOTA methods. Moreover, applying DPO fine-tuning further improves performance across all backbone configurations, demonstrating the generality and scalability of our approach. We will include these experimental results and analysis in the revised version.
>
> | Method| Backbone| NC| DAC  | EP| TTC | Comf | PDMS |
> | -------------- | ------------- | ---- | ---- | ---- | ---- | ----- | ---- |
> | GoalFlow| V2-99| 98.4 | 98.3 | 85.0 | 94.6 | 100.0 | 90.3 |
> | Hydra-MDP-B| V2-99| 98.4 | 97.8 | 86.5 | 93.9 | 100.0 | 90.3 |
> | Hydra-MDP-C| V2-99 & ViT-L | 98.7 | 98.2 | 86.5 | 95.0 | 100.0 | 91.0 |
> | Hydra-MDP++| V2-99| 98.6 | 98.6 | 85.7 | 95.1 | 100.0 | 91.0 |
> | Ours (w/o DPO) | V2-99| 98.6 | 98.7 | 84.7 | 95.0 | 99.9  | 90.5 |
> | Ours| V2-99| 98.9 | 99.1 | 85.2 | 95.9 | 100.0 | 91.4 |
>
> | Method|Backbone|NC| DAC| EP| TTC| Comf  | PDMS |
> | -------------- | -------- | ---- | ---- | ---- | ---- | ----- | ---- |
> | Hydra-MDP-A| ViT-L| 98.4 | 97.7 | 85.0 | 94.5 | 100.0 | 89.9 |
> | Ours (w/o DPO) | ViT-L| 98.2 | 98.1 | 84.1 | 94.0 | 100.0 | 89.6 |
> | Ours| ViT-L| 98.5 | 99.0 | 84.0 | 95.0 | 100.0 | 90.3 |
>
> [1] Jia, Xiaosong, et al. "Bench2drive: Towards multi-ability benchmarking of closed-loop end-to-end autonomous driving." NeurIPS 2024.
>
> [2] Li, Yongkang, et al. "ReCogDrive: A Reinforced Cognitive Framework for End-to-End Autonomous Driving." arXiv:2506.08052.
>
> [3] Lee, Youngwan, and Jongyoul Park. "Centermask: Real-time anchor-free instance segmentation." CVPR 2020.

---

### Official Review · Reviewer_VJCc · 2025-07-03

**Clarity:** 3
**Significance:** 2
**Originality:** 3
**Rating:** 4
**Confidence:** 4

**Summary:**

This paper proposes DriveDPO, a policy learning framework that brings the Direct Preference Optimization (DPO) method to end-to-end autonomous driving. To address the safety shortcomings of imitation learning (which can produce human-like but unsafe trajectories), the authors introduce a two-stage approach: (1) Unified Policy Distillation, which combines human imitation and rule-based safety scores to train an initial policy, and (2) Safety DPO, which fine-tunes this policy via pairwise preference learning on full trajectories. Experiments are conducted on the NAVSIM benchmark with a new metric (PDMS) and show gains over both imitation learning and rule-supervised baselines.

**Questions:**

1. How would the method perform if the rule-based safety scores (PDMS) were replaced by human preferences directly? Would a learned reward model generalize better?
2. Are the selection strategies for preference pairs (especially imitation-based) robust across datasets? How sensitive are results to the threshold?
3. Would the method handle out-of-distribution data, like in more complex or unseen scenarios?

**Ethical Concerns:**

["NO or VERY MINOR ethics concerns only"]

**Final Justification:**

The rebuttal properly addressed all my questions.

**Limitations:**

yes

**Quality:**

3

**Strengths And Weaknesses:**

Strengths:
1. Applying preference-based learning via DPO to autonomous driving is a logical and increasingly relevant extension of RLHF methods.
2. The idea of adapting DPO to control tasks is practical, leveraging recent successes in language modeling for robotics/control.
3. Performance on PDMS is solid, with gains over prior methods such as Hydra-MDP and WOTE, and qualitative results suggest better safety behavior.

Weaknesses:
1. The paper claims to distinguish between human-like but unsafe trajectories, yet does not provide empirical or qualitative evidence that the learned preferences actually capture safety-related behaviors. While the paper is motivated by this issue, the safety scoring function still heavily depends on rule-based metrics rather than explicitly learning to recognize subtle forms of human-like but risky behavior.
2. DriveDPO is a relatively straightforward combination of known components—policy distillation with hand-crafted safety scores, and standard DPO. The technical contribution lies more in integration than innovation.

---

> ### Author Rebuttal · Authors · 2025-07-30
>
> Thank you for your detailed and constructive reviews. We are glad that you found our method “is a logical and increasingly relevant extension of RLHF methods” and “performance on PDMS is solid”. We would like to address the Weaknesses (**W**) and Questions (**Q**) below.
>
> **[W1.1, Q1] The safety scoring function still heavily depends on rule-based metrics rather than explicitly learning to recognize subtle forms of human-like but risky behavior. Would a learned reward model generalize better?**
>
> Thank you for the insightful suggestion. We trained a reward model to distinguish between *human-like and safe* trajectories versus *human-like but unsafe* ones, and used it for DPO training. As shown in the table below, we found that replacing rule-based metrics with a learned reward model is effective. This may be because it can explicitly learn to recognize human-like but risky behaviors in ambiguous or complex scenarios that cannot be well captured by handcrafted rules, thus offering better generalization. We will include this result analysis in the revised version.
>
> | Method               | NC   | DAC  | EP   | TTC  | Comf  | PDMS |
> | -------------------- | ---- | ---- | ---- | ---- | ----- | ---- |
> | Baseline w/o DPO     | 97.9 | 97.3 | 84.0 | 93.6 | 100.0 | 88.8 |
> | Rule-based reward    | 98.4 | 97.5 | 83.5 | 94.6 | 100.0 | 89.3 |
> | Learned reward model | 98.2 | 98.2 | 83.7 | 94.3 | 100.0 | 89.6 |
>
>
> **[W1.2] The paper claims to distinguish between human-like but unsafe trajectories, yet does not provide empirical or qualitative evidence that the learned preferences actually capture safety-related behaviors.**
>
> To analyze the overall model preference, we computed the proportion of *human-like but unsafe* and *human-like and safe* trajectories among the Top 100 predicted trajectories on the test set (see the table below). After DPO fine-tuning, the proportion of *human-like but unsafe* trajectories significantly decreased, while the proportion of *human-like and safe* trajectories notably increased, indicating that the learned preferences can effectively distinguish such safety-related behaviors. Additionally, we present qualitative results in Figure 5 of the main paper, which further demonstrate the model’s improved ability to capture safety-related behaviors after DPO fine-tuning.
>
> | Method  | Human-like but unsafe rate (%) | Human-like and safe rate (%) |
> | :-----: | :----------------------------: | :--------------------------: |
> | w/o DPO |             38.44              |            61.56             |
> | w/ DPO  |             12.81              |            87.19             |
>
>
>
> **[W2] DriveDPO is a relatively straightforward combination of known components. The technical contribution lies more in integration than innovation.**
>
> We would like to respectfully clarify that our method goes beyond a simple integration of known components, but involves non-trivial innovations to introduce Safety DPO into end-to-end autonomous driving. First, instead of prior methods that independently predict multiple scores and implicitly derive a policy, we unify imitation similarity and rule-based safety into a single policy distribution that explicitly guides policy learning. Second, rather than naively selecting preference pairs based on safety scores,  we specifically address the human-like but unsafe issue in imitation learning by proposing the Imitation-Based Selection strategy, which focuses on more effective learning of safety-related preferences. The effectiveness of our method is consistently demonstrated across both the NAVSIM and Bench2Drive datasets, as well as under different vision backbones.
>
>
>
> **[Q1] How would the method perform if the rule-based safety scores (PDMS) were replaced by human preferences directly?**
>
> We directly replaced the rule-based scores with human preferences and observed that, while the performance improved compared to the baseline, it was still inferior to our proposed method (as shown in the Table below). This may be because of the absence of safety-oriented human preference data at the trajectory level, which means that even if the model aligns with human preferences, the absence of rule-based safety constraints may still lead to unsafe driving behaviors. However, we believe that with access to safety-relevant and fine-grained human preference datasets, which are currently lacking in the driving community, the model’s performance could be further improved. This will be an important direction we plan to explore in future work.
>
> | Method                 | NC   | DAC  | EP   | TTC  | Comf  | PDMS |
> | ---------------------- | ---- | ---- | ---- | ---- | ----- | ---- |
> | Baseline w/o DPO       | 97.9 | 97.3 | 84.0 | 93.6 | 100.0 | 88.8 |
> | Only human preferences | 98.0 | 97.9 | 84.1 | 93.7 | 100.0 | 89.3 |
> | Ours                   | 98.5 | 98.1 | 84.3 | 94.8 | 99.9  | 90.0 |
>
>
>
> **[Q2] Are the selection strategies for preference pairs (especially imitation-based) robust across datasets? How sensitive are results to the threshold?**
>
> As shown in the tables below, we conducted closed-loop evaluation experiments on the Bench2Drive dataset [1] and tested the Multi-Ability success rate across various complex driving scenarios. Our method achieves improvements over representative baselines in key metrics such as Driving Score, Success Rate, and Mean Multi-Ability Success Rate, demonstrating its robustness across datasets. In particular, our method achieves the highest success rate on the Emergency Brake task, indicating that the learned policy is more responsive to potentially risky behaviors.
>
> | Method | Efficiency | Comfortness | Success Rate(%) | Driving Score |
> | ------ | ---------- | ----------- | --------------- | ------------- |
> | AD-MLP | 48.45      | 22.63       | 0.00            | 18.05         |
> | UniAD  | 129.21     | 43.58       | 16.36           | 45.81         |
> | VAD    | 157.94     | 46.01       | 15.00           | 42.35         |
> | TCP    | 76.54      | 18.08       | 30.00           | 59.90         |
> | Ours   | 166.80     | 26.79       | 30.62           | 62.02         |
>
> | Method | Merging | Overtaking | Emergency Brake | Give Way | Traffic Sign | Mean  |
> | ------ | ------- | ---------- | --------------- | -------- | ------------ | ----- |
> | AD-MLP | 0.00    | 0.00       | 0.00            | 0.00     | 4.35         | 0.87  |
> | UniAD  | 14.10   | 17.78      | 21.67           | 10.00    | 14.21        | 15.55 |
> | VAD    | 8.11    | 24.44      | 18.64           | 20.00    | 19.15        | 18.07 |
> | TCP    | 8.89    | 24.29      | 51.67           | 40.00    | 46.28        | 34.22 |
> | Ours   | 16.28   | 28.95      | 53.06           | 30.00    | 45.00        | 34.66 |
>
> We also conducted ablation studies on preference pair selection strategies using the Bench2Drive dataset. As shown below, all proposed strategies lead to performance improvements in closed-loop evaluation, confirming the effectiveness and robustness of our design.
>
> | Method                              | Success Rate(%) | Driving Score |
> | ----------------------------------- | --------------- | ------------- |
> | Ours (vanilla Selection)            | 29.95           | 61.39         |
> | Ours (w/ Distance-Based Selection)  | 30.48           | 61.72         |
> | Ours (w/ Imitation-Based Selection) | 30.62           | 62.02         |
>
> We further performed ablation on the threshold sensitivity of the selection strategy (as shown below). Results show that under all threshold values, the performance remain higher than the baseline, suggesting robustness to threshold variations. We finally adopted $\tau = 0.3$ as the default setting based on overall performance.
>
> | threshold | NC   | DAC  | EP   | TTC  | Comf  | PDMS |
> | --------- | ---- | ---- | ---- | ---- | ----- | ---- |
> | w/o DPO   | 97.9 | 97.3 | 84.0 | 93.6 | 100.0 | 88.8 |
> | 0.1       | 98.3 | 97.8 | 83.5 | 94.4 | 99.9  | 89.4 |
> | 0.2       | 98.2 | 98.0 | 84.4 | 94.2 | 100.0 | 89.7 |
> | 0.3       | 98.5 | 98.1 | 84.3 | 94.8 | 99.9  | 90.0 |
> | 0.5       | 98.2 | 97.6 | 85.0 | 93.7 | 99.9  | 89.5 |
>
>
>
> **[Q3] Would the method handle out-of-distribution data, like in more complex or unseen scenarios?**
>
> As shown in the table below, we directly tested the trained model on the NuPlan dataset [2], which includes complex scenarios such as Heavy Traffic and Unprotected Cross Turn, as well as out-of-distribution settings where the city is changed to Boston. The model still performs well across all metrics, demonstrating the robustness and generalization ability of our method on out-of-distribution data.
>
> | Scenarios in Nuplan    | NC   | DAC  | EP   | TTC  | Comf  | PDMS |
> | ---------------------- | ---- | ---- | ---- | ---- | ----- | ---- |
> | Heavy Traffic          | 98.3 | 98.4 | 86.5 | 94.0 | 99.9  | 90.7 |
> | Unprotected Cross Turn | 98.7 | 97.9 | 82.4 | 95.5 | 99.9  | 89.5 |
> | In City Boston         | 97.5 | 97.0 | 83.2 | 92.0 | 100.0 | 87.6 |
>
>
>
> [1] Jia, Xiaosong, et al. "Bench2drive: Towards multi-ability benchmarking of closed-loop end-to-end autonomous driving." NeurIPS 2024.
>
> [2] Caesar, Holger, et al. "nuplan: A closed-loop ml-based planning benchmark for autonomous vehicles." arXiv:2106.11810.

---

> > ### Comment · Reviewer_VJCc · 2025-08-05
> >
> > Thanks for addressing my concern. I've updated my rating.

---

> > > ### Author Response · Authors · 2025-08-05
> > >
> > > Thank you sincerely for updating your rating. We truly appreciate your recognition of our work and the opportunity to further improve it through your valuable comments.

---

### Note · Authors · 2025-08-11

We sincerely thank the AC and all reviewers for their thoughtful evaluation and constructive suggestions. We are encouraged that our work is recognized as introducing an interesting and valuable direction for the community, with potential relevance for a wide range of researchers working on policy learning and autonomous driving (o9SK). We also appreciate the recognition of our methodology as a logical and timely extension of RLHF to end-to-end driving (VJCc), as well as the acknowledgment of our efforts to achieve meaningful gains on safety-related metrics (2wBa). Furthermore, we are pleased that the paper is considered clearly motivated, well-written, and easy to follow (ndFx). We have carefully addressed each comment from the reviewers and believe that most of their concerns have been effectively resolved.

We summarize the core contributions of our work as follows:

1. **Analysis of existing paradigm limitations**: We identify that existing imitation learning methods may generate trajectories that appear human-like yet violate safety principles. Moreover, score-learning methods learn trajectory scores independently and fail to explicitly optimize the policy.

2. **A novel policy learning framework**: We propose DriveDPO, a two-stage framework that first unifies rule-based and imitation-based supervision through policy distribution distillation, followed by trajectory-level iterative DPO using safety-aligned preferences. This enables explicit alignment of policies toward behaviors that are both safe and human-like.

3. **SOTA performance on challenging datasets**: Our method achieves SOTA results on both the open-loop NAVSIM and closed-loop Bench2Drive datasets. Notably, DriveDPO significantly improves safety-critical metrics compared with prior methods.

Finally, we would like to express our great appreciation and excitement that several reviewers regarded our framework as valuable to the driving community and as a timely direction that bridges RLHF and autonomous driving. We are excited by the potential of DriveDPO to raise broader community awareness of the safety limitations of imitation learning and inspire further exploration of safety-enhancing approaches.

---

### Decision · Program_Chairs · 2025-09-17

**Decision:**

Accept (poster)

**Comment:**

This paper proposes a two-stage policy learning framework called DriveDPO for end-to-end autonomous driving. The proposed framework uses both imitation learning and rule-based safety signals and refines the policy using DPO. The overall framework achieves good performance on the NAVSIM benchmark, with noticeable improvement on safety-related metrics.

The paper received generally positive feedback. The reviewers acknowledged its logical design and the performance of DriveDPO on the NAVSIM benchmark. There are also some concerns regarding this paper, specifically
- The novelty of the paper seems limited. It is built upon the Transfuser backbone, anchor vocabulary from VADv2, and the use of DPO and and rule-based score with imitation learning score is also trivial.
- The performance improvement does not seem to be very impressive. The proposed method uses additional rule-based information for policy learning, while many other baselines do not use such extra information, making the comparison not that fair for other baselines. Its performance suffers from a noticeable drop when removing rule-based signals (Table 2). Moreover, the improvement from adding DPO also seems pretty marginal (Table 1).

I encourage the authors to include the closed-loop evaluation results as well as some of the discussions during rebuttal in the final version of the paper.